# PARSED CATEGORIC ENCODINGS WITH AUTOMUNGE

## ABSTRACT

The Automunge open source python library platform for tabular data pre-processing automates feature engineering data transformations of numerical encoding and missing data infill to received tidy data on bases fit to properties of columns in a designated train set for consistent and efficient application to subsequent data pipelines such as for inference, where transformations may be applied to distinct columns in "family tree" sets with generations and branches of derivations. Included in the library of transformations are methods to extract structure from bounded categorical string sets by way of automated string parsing, in which comparisons between entries in the set of unique values are parsed to identify character subset overlaps which may be encoded by appended columns of boolean overlap detection activations or by replacing string entries with identified overlap partitions. Further string parsing options, which may also be applied to unbounded categoric sets, include extraction of numeric substring partitions from entries or search functions to identify presence of specified substring partitions. The aggregation of these methods into "family tree" sets of transformations are demonstrated for use to automatically extract structure from categoric string compositions in relation to the set of entries in a column, such as may be applied to prepare categoric string set encodings for machine learning without human intervention.

## 1 AUTOMUNGE

Automunge is an open source python library, available now for pip install, built on top of Pandas (McKinney, 2010), SciKit-Learn (Pedregosa et al., 2011), Scipy (Virtanen et al., 2020), and Numpy (van der Walt et al., 2011). It takes as input tabular data received in a tidy form (Wickham, 2014), meaning one column per feature and one row per observation, and returns numerically encoded sets with infill to missing points, thus providing a push-button means to feed raw tabular data directly to machine learning algorithms. The complexity of numerical encodings may be minimal, such as automated normalization of numerical sets and encoding of categorical, or may include more elaborate feature engineering transformations applied to distinct columns. Generally speaking, the transformations are performed based on a "fit" to properties of a column in a designated train set (e.g. based on a set's mean, standard deviation, or categorical entries), and then that same basis is used to consistently and efficiently apply transformations to subsequent designated test sets, such as may be intended for use in inference or for additional training data preparation.

The library consists of two master functions, automunge(.) and postmunge(.). The automunge(.) function receives a raw train set and if available also a consistently formatted test set, and returns a collection of encoded sets intended for training, validation, and inference. The function also returns a populated python dictionary, which we call the postprocess_dict, capturing all of the steps and parameters of transformations. This dictionary may then be passed along with subsequent test data to the postmunge(.) function for consistent processing on the train set basis, such as for instance may be applied sequentially to streams of data for inference. Because it makes use of train set properties evaluated during a corresponding automunge(.) call instead of directly evaluating properties of the test data, processing of subsequent test data in the postmunge(.) function is very efficient.

Included in the platform is a library of feature engineering methods, which in some cases may be aggregated into sets to be applied to distinct columns. For such sets of transformations, as may include generations and branches of derivations, the order of implementation is designated by passing transformation categories as entries to a set of "family tree" primitives described further below.

## 2 CATEGORIC ENCODINGS

In tabular data applications, such as are the intended use for Automunge data preparations, a common tactic for practitioners is to treat categoric sets in a coarse-grained representation for presentation to a training operation. Such aggregations transform each unique categoric entry with distinct numeric encoding constructs, such as one-hot encoding in which unique entries are each represented with their own column for boolean activations, or ordinal encoding of single column integer representations. The Automunge library offers some further variations on categoric encodings [Fig. 1]. The default ordinal encoding 'ord3', activated when the number of unique entries exceeds some heuristic threshold, has encoding integers sorted by frequency of occurrence followed by alphabetical, where frequency in general may be considered more useful to a training operation. For sets with a number of unique entries below this threshold, the categoric defaults instead make use of '1010' binary encodings, meaning multi-column boolean activations in which representations of distinct categoric entries may be achieved with multiple simultaneous activations, which has benefits over one-hot encoding of reduced memory bandwidth. Special cases are made for categoric sets with 2 unique entries, which are converted by 'bnry' to a single column boolean representation, and sets with 3 unique entries are somewhat arbitrarily applied with 'text' one-hot encoding. Note that each transform has a default convention for infill to missing values which may be updated for configure to distinct columns.

| column1 | 'text' (one-hot encoding) | | | | '1010' (binary encoding) | | |
|---|---|---|---|---|---|---|---|
| | column1_1234 | column1_circle | column1_square | column1_triangle | column1_1010_0 | column1_1010_1 | column1_1010_2 |
| circle | 0 | 1 | 0 | 0 | 0 | 0 | 1 |
| circle | 0 | 1 | 0 | 0 | 0 | 0 | 1 |
| circle | 0 | 1 | 0 | 0 | 0 | 0 | 1 |
| square | 0 | 0 | 1 | 0 | 0 | 1 | 0 |
| square | 0 | 0 | 1 | 0 | 0 | 1 | 0 |
| triangle | 0 | 0 | 0 | 1 | 0 | 1 | 1 |
| 1234 | 1 | 0 | 0 | 0 | 0 | 0 | 0 |
| NaN | 0 | 0 | 0 | 0 | 1 | 0 | 0 |
| NaN | 0 | 0 | 0 | 0 | 1 | 0 | 0 |

| column1 | 'ordl' (ordinal) | 'ord3' (ordinal by frequency) | | column2 | 'bnry' (boolean) |
|---|---|---|---|---|---|
| | column1_ordl | column1_ord3 | | | column2_bnry |
| circle | 1 | 0 | | yes | 1 |
| circle | 1 | 0 | | yes | 1 |
| circle | 1 | 0 | | yes | 1 |
| square | 2 | 1 | | yes | 1 |
| square | 2 | 1 | | no | 0 |
| triangle | 3 | 4 | | no | 0 |
| 1234 | 0 | 3 | | no | 0 |
| NaN | 4 | 2 | | NaN | 1 |
| NaN | 4 | 2 | | NaN | 1 |

Figure 1: Categoric encoding examples

The categoric encoding defaults in general are based on assumptions of training models in the decision tree paradigms (e.g. Random Forest, Gradient Boosting, etc.), particularly considering that a binary representation may sacrifice compatibility for an entity embedding of categorical variables [4] as may be applied in the context of a neural network (for which we recommend a seeded representation of 'ord3' for ordinal encoding by frequency). The defaults for automation are all configurable, and alternate root categories of transformations may also be specified to distinct columns to overwrite the automation defaults.

The characterization of these encodings as a coarse graining is meant to elude to the full discarding of the structure of the received representations. When we convert a set of unique values {'circle', 'square', 'triangle'} to {1, 2, 3}, we are hiding from the training operation any information that might be inferred from grammatical structure, such as the recognition of the prefix "tri" meaning three. Consider the word "Automunge". You can probably infer from the common prefix "auto" that there might be some automation involved, or if you are a data scientist you might recognize

the word "munge" as referring to data transformations. Naturally we may then ask how we might incorporate practices from NLP into our tabular encodings, such as vectorized representations in a model like Word2Vec (Mikolov et al., 2013). While this is a worthy line for further investigation, some caution is deserved that the validity of such representations is built on a few assumptions, most notably the consistency of vocabulary interpretations between any pre-trained model and the target tabular application. In NLP practice it is common to fine-tune (Howard & Ruder, 2018) a pre-trained model to accommodate variation in a target domain. In tabular applications obstacles to this type of approach may arise from the limited context surrounding the categoric entries - which in practice is not uncommon to find entries as single words, or sometimes character sets that aren't even words such as e.g. serial numbers or addresses. We may thus be left without the surrounding corpus vocabulary necessary to fine tune a NLP model for some tabular target, and thus the only context available to extract domain properties may be the other entries shared in a categoric set or otherwise just the surrounding corresponding tabular features.

## 3 STRING PARSING

Automunge offers a novel means to extract some additional grammatical context from categoric entries prior to encoding by way of comparisons between entries in the unique values of a feature set. The operation is conducted by a kind of string parsing, in which the set of unique entries in a column are parsed to identify character subset overlaps between entries. In its current form the implementation is probably not at an optimal computational efficiency, which we estimate complexity scaling on the order of $\mathcal{O}((LN)^2)$, where N is the number of unique entries and L is the average character length of those entries. Thus the intended use case is for categoric sets with a bounded range of unique entries. In general, the application of comparable transformations to test data in the postmunge(.) function is materially more computationally efficient than the initial fitting of the transformations to the train set in the automunge(.) function, particularly in variations with added assumptions for test set composition in relation to the corresponding train data.

The implementation to identify character subset overlaps is performed by first inspecting the set of unique entries from the train set, determining the longest string length (-1), then for each entry comparing each subset of that length to every equivalent length subset of the other entries, where if composition overlaps are identified, the results are populated in a data structure matching identified overlaps to their corresponding source column unique entries, and after each overlap inspection cycle incrementing that inspection length by negative steps until a configurable minimum length overlap detection threshold is reached. To keep things manageable, in the base configuration overlaps are limited to a single identification per unique entry, prioritized by longest length. Note also that transformation function parameters may be activated to exclude from overlap detections any subsets with space, punctuation, or other designated character sets such as to promote single word activations.

There are a few options of these methods to choose from [Fig. 2]. In the first version 'splt' any identified substring overlap partitions are given unique columns for boolean activations. In a second version 'spl2' the full entries with identified overlaps are replaced with the overlap partitions, resulting in a reduced number of unique entries - such as may be intended as a reduced information content supplement to the original set, which we speculate could be beneficial in the context of a curriculum learning regime (Bengio et al., 2009). A third version 'spl5' is similar to the second with distinction that those entries not replaced with identified overlap partitions are instead replaced with an infill plug value, such as to avoid too much redundancy between different configurations derived from the same set. The fourth shown version 'sp15' is comparable to 'splt' but with the allowance for multiple concurrent activations to each entry, demonstrating a tradeoff between information retention and dimensionality of returned data. Each of these versions have corresponding variants with improved computational efficiency to test set processing with the postmunge(.) function based on incorporating the assumption that the set of unique entries in the test set will be the same or a subset of those found in the train set.

Some further variations include 'sp19' in which concurrent activation sets are collectively consolidated with a binary transform to reduce dimensionality. Another variation is available as 'sbst' which instead of comparing string character subsets of entries to string character subsets of other entries, only compares string character subsets of entries to complete character sets of other entries.

| address | 'splt' (string overlap identification) | | | 'spl2' (string overlap ordinal) | 'spl5' (exclude non-overlaps) |
|---|---|---|---|---|---|
| | address_splt_South Peterson St Maitland, FL 32789 | address_splt_North Peterson St Orlando, FL 32714 | address_splt_St Altamonte Springs, FL 32715 | address_spl2_ord3 | address_spl5_ord3 |
| 1234 North Peterson St Orlando, FL 32714 | 0 | 1 | 0 | 0 | 1 |
| 2345 South Anderson St Altamonte Springs, FL 32715 | 0 | 0 | 1 | 2 | 3 |
| 3456 South Peterson St Maitland, FL 32789 | 1 | 0 | 0 | 1 | 2 |
| 4567 North Peterson St Orlando, FL 32714 | 0 | 1 | 0 | 0 | 1 |
| 5678 Avenue St Orlando, FL 32714 | 0 | 0 | 0 | 3 | 0 |
| 6789 South Peterson St Maitland, FL 32789 | 1 | 0 | 0 | 1 | 2 |
| 5858 North Other St Altamonte Springs, FL 32715 | 0 | 0 | 1 | 2 | 3 |
| None | 0 | 0 | 0 | 4 | 0 |
| Orlando, FL | 0 | 0 | 0 | 5 | 0 |

| address | 'sp15' string overlap with allowed concurrent activations | | | | | | | |
|---|---|---|---|---|---|---|---|---|
| | address_sp15_South Peterson St Maitland, FL 32789 | address_sp15_North Peterson St Orlando, FL 32714 | address_sp15_St Altamonte Springs, FL 32715 | address_sp15_St Orlando, FL 32714 | address_sp15_Orlando, FL | address_sp15_erson St | address_sp15_South | address_sp15_North |
| 1234 North Peterson St Orlando, FL 32714 | 0 | 1 | 0 | 1 | 1 | 1 | 0 | 1 |
| 2345 South Anderson St Altamonte Springs, FL 32715 | 0 | 0 | 1 | 0 | 0 | 1 | 1 | 0 |
| 3456 South Peterson St Maitland, FL 32789 | 1 | 0 | 0 | 0 | 0 | 1 | 1 | 0 |
| 4567 North Peterson St Orlando, FL 32714 | 0 | 1 | 0 | 1 | 1 | 1 | 0 | 1 |
| 5678 Avenue St Orlando, FL 32714 | 0 | 0 | 0 | 1 | 1 | 0 | 0 | 0 |
| 6789 South Peterson St Maitland, FL 32789 | 1 | 0 | 0 | 0 | 0 | 1 | 1 | 0 |
| 5858 North Other St Altamonte Springs, FL 32715 | 0 | 0 | 1 | 0 | 0 | 0 | 0 | 1 |
| None | 0 | 0 | 0 | 0 | 0 | 0 | 0 | 0 |
| Orlando, FL | 0 | 0 | 0 | 0 | 1 | 0 | 0 | 0 |

** splt / spl2 / spl5 / sp15 have comparable variants spl8 / spl9 / sp10 / sp16 with added assumptions of test set composition for efficiency.

Figure 2: String parsing for bounded sets

## 4 PARSING UNBOUNDED SETS

The transformations of the preceding section were somewhat constrained toward use against categoric sets with a bounded range of unique entries in the training data due to the complexity scaling on train set implementation. For cases where categoric sets may be presented with an unbounded range of unique entries, such as in cases for all unique entries or otherwise unbounded, Automunge offers a few more string parsing transformations to extract grammatical structure prior to categoric encodings.

As a first example, categoric entries may be passed to a search function 'srch' in which entries are string parsed to identify presence of user-specified substring partitions, which are then presented to machine learning by way of recorded activations in returned boolean columns specific to each search term or alternatively with the set of activations collected into a single ordinal encoded column. Some variations on the search functions include the allowance to aggregate multiple search terms into common activations or variations for improved processing efficiency based on added assumptions of whether the target set has a narrow range of entries. Note that these methods are supported by the Automunge infrastructure allowing a user to pass transformation function specific parameters to those applied to distinct columns or also to reset transformation function parameter defaults in the context of an automunge(.) call.

Another string parsing option suitable for application to both bounded and unbounded categoric sets is intended for purposes of detecting numeric portions of categoric string entries, which are extracted and returned in a dedicated numeric column. Some different numeric formats are supported, including entries with commas via 'nmcm', and using the family tree primitives the returned sets may in the same operation be normalized such as with a z-score or min-max scaling. As with other string-parsing methods priority of extracts are given to identified partitions with the longest character length. A comparable operation may instead be performed to extract string partitions from majority numeric entries, although we suggest caution of applying these methods towards numerical sets which include units of measurement for instance, as we recommend reserving engineering domain evaluations for oversight by human intelligence.

| address | 'nmcm' (string parse for number, commas ok) address_nmcm | 'nmc2' (nmcm with z-score) address_nmcm_nmbr | 'nmc3' (nmcm with min-max) address_nmcm_mnmx | 'srch' categoric string search with passed 'search' parameter: ['Maitland', 'Orlando', 'Altamonte Springs'] address_srch_Maitland | | address_srch_Altamonte Springs | 'src4' (comparable to srch, ordinal) address_src4 |
|---|---|---|---|---|---|---|---|
| | | | | | address_srch_Orlando | | |
| 1234 North Peterson St Orlando, FL 32714 | 32714.000000 | -0.688760 | 0.000000 | 0 | 1 | 0 | 2 |
| 2345 South Anderson St Altamonte Springs, FL 32715 | 32715.000000 | -0.657041 | 0.013333 | 0 | 0 | 1 | 3 |
| 3456 South Peterson St Maitland, FL 32789 | 32789.000000 | 1.690181 | 1.000000 | 1 | 0 | 0 | 1 |
| 4567 North Peterson St Orlando, FL 32714 | 32714.000000 | -0.688760 | 0.000000 | 0 | 1 | 0 | 2 |
| 5678 Avenue St Orlando, FL 32714 | 32714.000000 | -0.688760 | 0.000000 | 0 | 1 | 0 | 2 |
| 6789 South Peterson St Maitland, FL 32789 | 32789.000000 | 1.690181 | 1.000000 | 1 | 0 | 0 | 1 |
| 5858 North Other St Altamonte Springs, FL 32715 | 32715.000000 | -0.657041 | 0.013333 | 0 | 0 | 1 | 3 |
| None | 32735.714844 | 0.000000 | 0.289524 | 0 | 0 | 0 | 0 |
| Orlando, FL | 32735.714844 | 0.000000 | 0.289524 | 0 | 1 | 0 | 2 |

Figure 3: Numeric extraction and search for unbounded sets

# 5 FAMILY TREE AGGREGATIONS

An example composition of string parsing transformation aggregations, including generations and branches of derivations by way of entries to the family tree primitives, are now demonstrated for the root transformation category 'or19' [Fig. 4], which is available for assignment to source columns in context of an automunge(.) call. This transformation set is intended to automatically extract grammatical context from tabular data categoric features with a bounded range of unique entries.

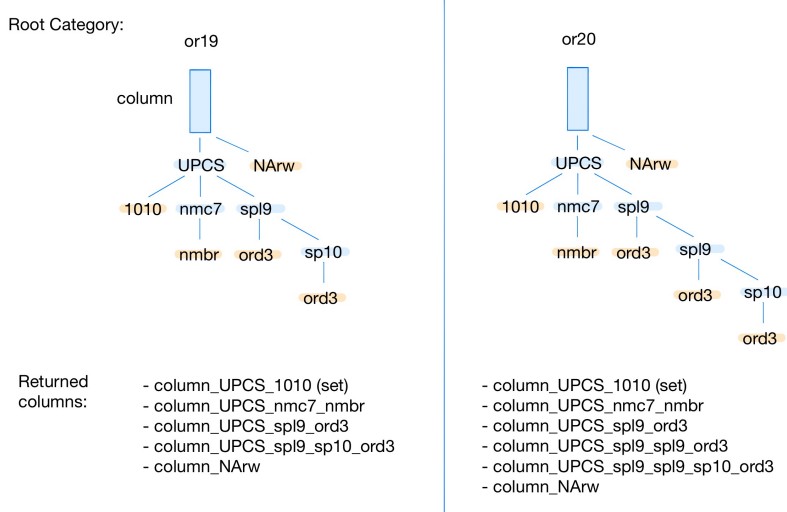

Figure 4: Example family tree aggregations for bounded categoric sets

The sequence of four character keys represent transformation functions based on the transformation categories applied to a column. Note that each key represents a set of functions which may include one for application to train/test set(s) in an automunge(.) call for initial fitting of transformations to properties of a train set or a corresponding function for processing of a comparably formatted test set in postmunge(.) on the basis of properties from the train set. The steps of transformations for each returned column are logged by way of transformation function specific suffix appenders to the original column headers. Note also that some of the intermediate steps of transformations may not be retained in the returned set based on presence of downstream replacement primitive entries to family tree primitives as described further below.

The upstream application of an 'UPCS' transform serves the purpose of converting categoric entry strings to all uppercase characters, thus consolidating entries with different case configurations, e.g. a received set with unique entry strings 'usa', 'Usa', 'USA' would all be considered an equivalent entry (there may be some domains where this convention is not preferred in which case a user may

deactivate a parameter to exclude this step). Adjacent to the 'UPCS' transform is a 'NArw' which returns boolean activations for those rows corresponding to infill based on the root category "processdict" defined type of source column values that will be target for infill (processdict is a data structure discussed further below, whose entries may either be a default or user specified). The included '1010' transform for binary encoding distinguishes all distinct entries prior to any string parsing aggregations to ensure full information retention after the 'UPCS' transform both for purposes of ML training and also to support any later inversion operation to recover the original format of data pre-transforms as is supported in the postmunge(.) function. An alternate configuration could replace '1010' with another categoric transform such as one-hot or ordinal encodings. The 'nmc7' transformation function is similar to 'nmcm' discussed earlier, but only string parses those unique entries which were not found in the train set for a more efficient application in the postmunge(.) function. These numeric extractions are followed by a 'nmbr' function for z-score normalization. Note that in some cases a numerical extract, such as those derived here from zip codes of passed addresses, may in fact be more suitable to a categoric encoding instead of numeric normalization. Such alternate configurations may easily be defined with the family tree primitives.

The remaining branches downstream of the 'UPCS' start with a 'spl9' function performing string parsing replacement operations comparable to the 'spl2' demonstrated above, but with the assumption that the set of unique entries in the test data will be the same or a subset of the train set for efficiency considerations. The 'spl9' parses through unique entry strings to identify character subset overlaps and replaces entries with the longest identified overlap, which results in returned columns with a fewer number of unique entries by aggregating entries with shared overlaps into a common representation, which may then be numerically encoded such as shown here with an 'ord3' transform. The 'or19' family tree also incorporates a second tier of string parsing with overlap replacement by use of the 'sp10' transform (comparable to 'spl5' with similar added assumptions for efficiency considerations as 'spl9'). 'sp10' differs from 'spl9' in that unique entries without overlap are replaced in aggregate with an infill plug value to avoid unnecessary redundancy between encodings, again resulting in a reduced number of unique entries which may then be numerically encoded such as with 'ord3'. Note that Fig. 4 also demonstrates an alternate root category configuration as 'or20' in which an additional tier of 'spl9' is incorporated prior to the 'sp10'.

| | 'or19' a "family tree" of transformations intended to extract grammatical context from bounded categoric sets of unknown composition | | | | | | | |
|---|---|---|---|---|---|---|---|
| **address** | address_UPCS_nmc7_nmbr | address_UPCS_1010_0 | address_UPCS_1010_1 | address_UPCS_1010_2 | address_UPCS_1010_3 | address_UPCS_spl9_ord3 | address_UPCS_spl9_sp10_ord3 | address_NArw |
| 1234 North Peterson St Orlando, FL 32714 | -0.688760 | 0 | 0 | 0 | 0 | 0 | 1 | 0 |
| 2345 South Anderson St Altamonte Springs, FL 32715 | -0.657041 | 0 | 0 | 0 | 1 | 2 | 0 | 0 |
| 3456 South Peterson St Maitland, FL 32789 | 1.690181 | 0 | 0 | 1 | 0 | 1 | 2 | 0 |
| 4567 North Peterson St Orlando, FL 32714 | -0.688760 | 0 | 0 | 1 | 1 | 0 | 1 | 0 |
| 5678 Avenue St Orlando, FL 32714 | -0.688760 | 0 | 1 | 0 | 0 | 3 | 1 | 0 |
| 6789 South Peterson St Maitland, FL 32789 | 1.690181 | 0 | 1 | 1 | 0 | 1 | 2 | 0 |
| 5858 North Other St Altamonte Springs, FL 32715 | -0.657041 | 0 | 1 | 0 | 1 | 2 | 0 | 0 |
| None | 0.000000 | 1 | 0 | 0 | 0 | 4 | 0 | 1 |
| Orlando, FL | 0.000000 | 0 | 1 | 1 | 1 | 5 | 0 | 0 |

Figure 5: Demonstration of 'or19' returned data

Fig. 5 demonstrates the numerical encodings as would be returned from the application of the 'or19' root category to a small example feature set of categoric strings. It might be worth restating that due to the complexity scaling of the string parsing operation this type of operation is intended preferably for categoric sets with a bounded range of unique entries in the train set. The composition of returned sets are derived based on properties of the source column received in a designated train set, and these same bases are applied to consistently prepare data for test sets, such as sets that may be intended for an inference operation. In other words, when preparing corresponding test data, the same type and order of columns are returned, with equivalent encodings for corresponding entries and equivalent activations for specific string subset overlap partitions that were found in the train set.

## 6 SPECIFICATION

The specification of transformation set compositions for these methods are conducted by way of transformation category entries to a set of family tree primitives, which distinguish for each transformation category the upstream transformation categories for when that category is applied as a root category to a source column and also the downstream transformation categories for when that category is found as an entry in an upstream primitive with offspring. Downstream primitive entries are treated as upstream primitive entries for the successive generation, and primitives further distinguish source column retention and generation of offspring.

| primitive | upstream / downstream | applied to generation | column action | downstream offspring |
|---|---|---|---|---|
| parents | upstream | first | replace | yes |
| siblings | upstream | first | supplement | yes |
| auntsuncles | upstream | first | replace | no |
| cousins | upstream | first | supplement | no |
| children | downstream parents | offspring | replace | yes |
| niecesnephews | downstream siblings | offspring | supplement | yes |
| coworkers | downstream auntsuncles | offspring | replace | no |
| friends | downstream cousins | offspring | supplement | no |

Figure 6: Family tree primitives

Transformation category family tree sets may be passed to an automunge(.) call by way of a "transformdict" data structure, which is complemented by a second "processdict" data structure populated for each transformation category containing entries for the associated transformation functions and data properties. The transformdict with transformation category entries to family tree primitives and corresponding processdict transformation function entries associated with various columns returned from the 'or19' root category set are demonstrated here [Fig. 7]. Here the single processdict entry of the transformation function associated with a transformation category is an abstraction for the set of corresponding transformation functions to be directed at train and/or test set feature sets.

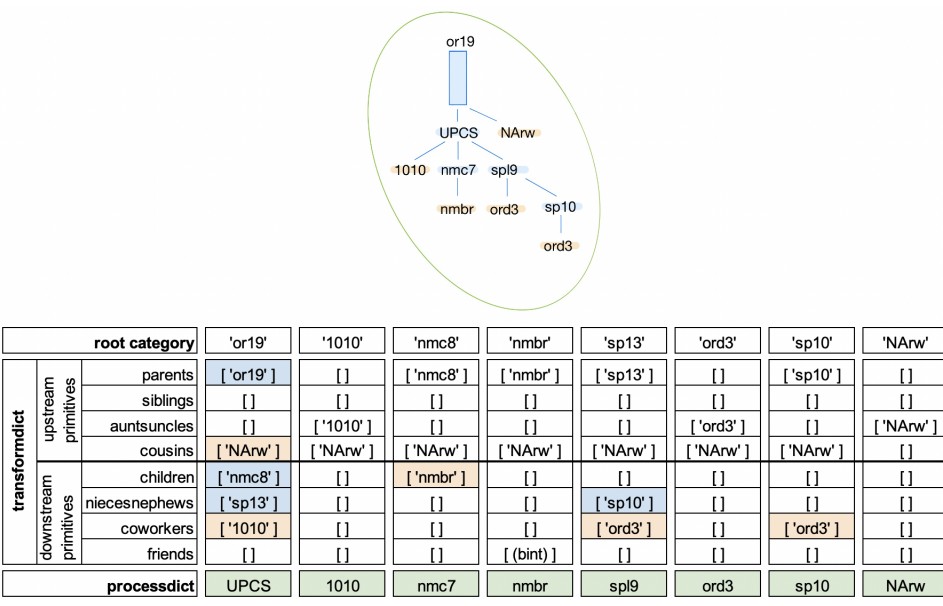

Figure 7: 'or19' specifications

## 7 EXPERIMENTS

Some experiments were run to evaluate comparison between standard tabular categoric representation techniques and parsed categoric encodings. The data set from the IEEE-CIS Kaggle competition (Vesta, 2019) was selected based on known instances of feature sets containing serial number entries which were expected as a good candidate for string parsing.

The experiments were supported by the Automunge library's feature importance evaluation methods, in which a model is trained on the full feature set to determine a base accuracy on a partitioned validation set, and metrics are calculated by shuffle permutation (Parr et al., 2018), where the target feature has it's entries shuffled between rows to measure damping effect on the resulting accuracy, the delta of which may serve as a metric for feature importance. Automunge actually aggregates two metrics for feature importance, the first metric specific to a source feature by shuffling all columns derived from the same feature, in which a higher metric represents more importance, and the second metric specific to each derived column by shuffling all but the target of the columns derived form the same feature, in which a lower metric represents more relative importance between columns derived from the same feature. The experiment findings discussed below are based on the first metric. Although other auto ML options are supported in the library, this experiment used the base configuration of Random Forest (Breiman, 2001) for the model.

For the experiment, the training data was paired down to the top ten features based on feature importance in addition to two features selected as targets for the experiments, identified in the data set by 'id_30' and 'id_31'. These two features contained entries corresponding to operating system serial numbers and browser serial numbers associated with origination of financial transactions. By inspection, there were many cases where serial numbers shared portions of grammatical structure, as for example the entries {'Mac OS X 10_11_6', 'Mac OS X 10_7_5'} or {'chrome 62.0', 'chrome 49.0'}. Scenarios were run in which both of these features were treated to different types of encodings, including 'text' one-hot encoding, 'ord3' ordinal, '1010' binary [Fig 1], and two string parse scenarios, the first with 'or19' [Fig 4, 5, 7] and the second with an aggregation of: 'sp19' (string parse with concurrent activations consolidated by binary encoding) supplemented by 'nmcm' (numeric extraction) [Fig 3] and 'ord3' (ordinal encoding). The feature importance was then evaluated corresponding to each of these encoding scenarios [Table 1].

Table 1: Feature Importance Metric Results

| Encoding | Category | Accuracy | 'id_30' | 'id_31' |
|---|---|---|---|---|
| one-hot | 'text' | 0.98029 | 0.00135 | 0.00490 |
| ordinal | 'ord3' | 0.98040 | 0.00193 | 0.00581 |
| binary | '1010' | 0.98045 | 0.00245 | 0.00699 |
| string parse | 'or19' | 0.98082 | 0.00295 | 0.00914 |
| string parse | 'sp19' | 0.98081 | 0.00279 | 0.00924 |

The experiments illustrate some important points about the impact of categoric encodings even outside of string parsing. Here we see that binary encoding materially outperforms one-hot encoding and ordinal encoding. We believe that one-hot encoding is best suited for labels or otherwise just used for interpretability purposes. We believe ordinal encoding is best suited for very high cardinality when a set has large number of entries. We believe binary is the best default for categoric encodings outside of vectorization, and thus serves as our categoric default under automation.

The string parsing was found to have a material benefit to both of our target features. It appears the 'or19' version of string parsing was more beneficial to the 'id_30' feature and the 'sp19' version to the 'id_31' feature.

Part of the challenge of benchmarking parsed categoric encodings is the nature of the application, in that performance impact of string parsing is highly dependent on data set properties, and not necessarily generalizable to a metric that would be relevant for comparison between different features or data sets. We believe this experiment has successfully demonstrated that string parsing has the potential to train better performing models in tabular applications based on improved model accuracy and feature importance in comparisons for these specific features.

## 8 Discussion

To be clear, we believe the family tree primitives [Fig 6] represent a scientifically novel framework, serving as a fundamental reframing of command line specification for multi-transform sets as may include generations and branches of derivations applied by recursion. They are built on assumptions of tidy data and that derivations are all downstream of a specific target feature set, and are well suited for the final data preprocessing steps applied prior to the application of machine learning in tabular applications. We consider these primitives a universal language for univariate data transformation specification and an improvement on mainstream practice.

Although this paper is being submitted under the subject of NLP, it should be noted that the string parsing methods as demonstrated are kind of a compromise from vocabulary vectorization, intended for tabular applications in esoteric domains with limited context or surrounding language such as could be used to fine-tune a pre-trained model, and thus not suitable for mainstream NLP models like BERT. We have attempted in this work a comprehensive overview of various permutations of string parsing that may be applied for scenarios excluding vectorization. That is not to say that a vectorization may not still be achievable - for instance each of the returned categoric encodings of varying information content returned from 'or19' could be fed as input to an entity embedding layer [4] when the returned sets are used to train a model.

Further, this paper is not just intended to propose theory and methods. Automunge is a downloadable software toolkit, and the methods demonstrated here are available now in the library for push-button operation. It really is just as simple as passing a dataframe and designating a root category of 'or19' to a target column. We believe the automation of string parsing for categoric encodings is a novel invention that will prove very useful for machine learning researchers and practitioners alike.

The value of the library extends well beyond string parsing. For instance, Automunge is an automated solution to missing data infill. In addition to the infill defaults for each transformation, a user can select for each column other infill options from the library, including "ML infill" in which column specific Random Forest models (Breiman, 2001) are trained from partitioned subsets of the training data to predict infill to train and test sets. For example, when ML infill is applied to the 'or19' set, each of the returned subsets will have their own trained infill model.

An important point of value is not just the transformations themselves, but the means by which they are applied between train and test sets. In a traditional numerical set normalization operation for instance, it is not uncommon that each of these sets is evaluated individually for a mean and standard deviation, which runs a risk of inconsistency of transformations between sets, or alternate methods to measure prior to validation set extraction runs the risk of data leakage between training and validation operations. In a postmunge(.) test set application, all of the transformation parameters are derived from corresponding columns in the train set passed to automunge(.) after partitioning validation sets, which in addition to solving these problems of inconsistency and data leakage, we speculate that at data center scale could have material benefit to computational overhead and associated carbon intensity of inference, perhaps also relevant to edge device battery constraints.

Another key point of value for this platform is simply put the reproducibility of data preprocessing. If a researcher wants to share their results for exact duplication to the same data or similar application to comparable data, all they have to do is publish the simple python dictionary returned from an automunge(.) call, and other researchers can then exactly duplicate. The same goes for archival of preprocessing experiments - a source data set need only be archived once, and every performed experiment remains accessible and reproducible with this simple python dictionary.

Beyond the core points of feature engineering and infill, the Automunge library contains several other push-button methods. The goal is to automate the full workflow for tabular data for the steps between receipt of tidy data and returned sets suitable for machine learning application. Some of the options include feature importance evaluation (by shuffle permutation (Breiman, 2001)), dimensionality reduction (including by means of PCA (Jolliffe & Cadima, 2016), feature importance, and binary encodings), preparation for oversampling in cases of label set class imbalance (Buda et al., 2017), evaluation of data distribution drift between initial train sets and subsequent test sets, and perhaps most importantly the simplest means for consistently and efficiently processing subsequent data with postmunge(.).

Oh, and once you try it out, please let us know.

ACKNOWLEDGMENTS

A thank you owed to: the Kaggle IEEE-CIS competition which helped me recognize the potential for string parsing. Mark Ryan who shared a comment in Deep Learning with Structured Data that was inspiration for the 'UPCS' transform. Thanks to Stack Overflow, Python, PyPI, GitHub, Colaboratory, Anaconda, and Jupyter. Special thanks to Scikit-Learn, Numpy, Scipy Stats, and Pandas.

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

APPENDIX

# A  BROADER IMPACT

The following discussions are somewhat speculative in nature. At the time of this writing Automunge has yet to establish what we would consider a substantial user base and there may be a bias towards optimism at play in how we have been proceeding, which we believe is our sole leverage of bias.

From an ethical standpoint, we believe the potential benefits of our platform far outweigh any negative aspects. What we noted in the discussions about the potential for reduced carbon intensity for machine learning at scale has been one of our guiding principles for design. Particularly we have sought to optimize the postmunge(.) function for speed, used as a proxy for computational efficiency. As a rule of thumb, processing times for equivalent data in the postmunge(.) function, such as could be applied to streams of data in inference, have shown to operate on the order of twice the speed of initial preparations in the automunge(.) function, although for some specific transforms like those implementing string parsing that advantage may be considerably higher. While the overhead may prevent achieving the speed of directly applying manually specified transformations to a dataframe, the postmunge(.) speed gets close to manual transformations with increasing data size.

We believe too that the impact to the machine learning community of a formalized open source standard to tabular data preprocessing could have material benefits to ensuring reproducibility of results. There for some time has been a gap between the wide range of open source frameworks for training neural networks in comparison to options for prerequisites of data pipelines. I found some validation for this point from the tone of the audience Q&A at a certain 2019 NeurIPS keynote presentation by the founder of a commercial data wrangling package. In fact it may be considered a potential negative impact of this research in the risk to commercial models of such vendors, as Automunge's GNU GPL 3.0 license coupled with patent pending status on the various inventions behind our library (including these string parsing methods, family tree primitives, ML infill, and etc.) will preclude commercial platforms offering comparable functionality. We expect that the benefits to the machine learning community in aggregate will far outweigh the potential commercial impacts to this narrow segment.

Further, benefits of automating machine learning derived infill to missing data may result in a material impact to the mainstream data science workflow. That old rule of thumb often thrown around about how 80% of a machine learning project is cleaning the data may need to be revised to a lower figure. We speculate that other options for generalized missing data infill, such as methods built around do-calculus (Mohan & Pearl, 2018), may have benefits for generality across multiple target labels in a data lake (the ML infill method predicts infill on a basis of a specific target feature set), but there may be trade-offs for computational overhead of implementation, not to mention simplicity.

Regarding consequences of system failure, it should be noted that Automunge is an industry agnostic toolset, with intention to establish users across a wide array of tabular data domains, potentially ranging from the trivial to mission critical. We recognize that with this exposure comes additional scrutiny and responsibility. Our development has been performed by a professional engineer and we have sought to approach validations, which has been an ongoing process, with a commensurate degree of rigor.

Our development has followed an incremental and one might say evolutionary approach to systems engineering, with frequent and sequential updates as we iteratively added functionality and transforms to the library within defined boundaries of the data science workflow. The intent has always been to transition to a more measured pace at such time as we may establish a more substantial user base.

# B  FUNCTION CALL DEMONSTRATIONS

Automunge is available for pip install:

```
pip install Automunge
```

Or to upgrade (we currently roll out upgrades fairly frequently):

```
pip install Automunge --upgrade
```

Once installed, run this in local session to initialize:

```
from Automunge import Automunger
am = Automunger.AutoMunge()
```

Then, assuming we want to prepare a train set df_train for ML, can apply default parameters as:

```
train, trainID, labels, \
validation1, validationID1, validationlabels1, \
validation2, validationID2, validationlabels2, \
test, testID, testlabels, \
labelsencoding_dict, finalcolumns_train, finalcolumns_test, \
featureimportance, postprocess_dict = \
am.automunge(df_train)
```

Note that if our df_train set included a labels column, we should designate the column header with the labels_column parameter. Or likewise we can designate any ID columns with the trainID_column parameter.

The returned postprocess_dict should be saved such as with pickle.

We can then consistently prepare subsequent test data df_test in postmunge(.):

```
test, testID, testlabels, \
labelsencoding_dict, postreports_dict \
= am.postmunge(postprocess_dict, df_test)
```

I find it helps to just copy and paste the full range of parameters for reference:

```
train, trainID, labels, \
validation1, validationID1, validationlabels1, \
validation2, validationID2, validationlabels2, \
test, testID, testlabels, \
labelsencoding_dict, finalcolumns_train, finalcolumns_test, \
featureimportance, postprocess_dict = \
am.automunge(df_train, df_test=False, labels_column=False,
 trainID_column=False, testID_column=False, valpercent1=.0,
 valpercent2=.0, floatprecision=32, shuffletrain=True,
 TrainLabelFreqLevel=False, powertransform=False,
 binstransform=False, MLinfill=False, infilliterate=1,
 randomseed=42, eval_ratio=.5, LabelSmoothing_train=False,
 LabelSmoothing_test=False, LabelSmoothing_val=False, LSfit=False,
 numbercategoryheuristic=63, pandasoutput=False,
 NArw_marker=False, featureselection=False, featurepct=1.0,
 featuremetric=0.0, featuremethod='default', Binary=False,
 PCAn_components=False, PCAexcl=[], excl_suffix=False,
 ML_cmnd = {'MLinfill_type':'default',
          'MLinfill_cmnd':{'RandomForestClassifier':{},
                          'RandomForestRegressor':{}},
          'PCA_type':'default', 'PCA_cmnd':{}},
```

```
assigncat = {
'nmbr':[], 'retn':[], 'mnmx':[], 'mean':[], 'MAD3':[], 'lgnm':[],
'bins':[], 'bsor':[], 'pwrs':[], 'pwr2':[], 'por2':[], 'bxcx':[],
'addd':[], 'sbtr':[], 'mltp':[], 'divd':[],
'log0':[], 'log1':[], 'logn':[], 'sqrt':[], 'rais':[], 'absl':[],
'bnwd':[], 'bnwK':[], 'bnwM':[], 'bnwo':[], 'bnKo':[], 'bnMo':[],
'bnep':[], 'bne7':[], 'bne9':[], 'bneo':[], 'bn7o':[], 'bn9o':[],
'bkt1':[], 'bkt2':[], 'bkt3':[], 'bkt4':[],
'nbr2':[], 'nbr3':[], 'MADn':[], 'MAD2':[], 'tlbn':[],
'mnm2':[], 'mnm3':[], 'mnm4':[], 'mnm5':[], 'mnm6':[],
'ntgr':[], 'ntg2':[], 'ntg3':[], 'mea2':[], 'mea3':[], 'bxc2':[],
'dxdt':[], 'd2dt':[], 'd3dt':[], 'dxd2':[], 'd2d2':[], 'd3d2':[],
'nmdx':[], 'nmd2':[], 'nmd3':[], 'mmdx':[], 'mmd2':[], 'mmd3':[],
'shft':[], 'shf2':[], 'shf3':[], 'shf4':[], 'shf7':[], 'shf8':[],
'bnry':[], 'onht':[], 'text':[], 'txt2':[], '1010':[], 'or10':[],
'ordl':[], 'ord2':[], 'ord3':[], 'ord4':[], 'om10':[], 'mmor':[],
'Unht':[], 'Utxt':[], 'Utx2':[], 'Uor3':[], 'Uor6':[], 'U101':[],
'splt':[], 'spl2':[], 'spl5':[], 'sp15':[], 'sp19':[], 'sbst':[],
'spl8':[], 'spl9':[], 'sp10':[], 'sp16':[], 'sp20':[], 'sbs2':[],
'srch':[], 'src2':[], 'src4':[], 'strn':[], 'lngt':[], 'aggt':[],
'nmrc':[], 'nmr2':[], 'nmcm':[], 'nmc2':[], 'nmEU':[], 'nmE2':[],
'nmr7':[], 'nmr8':[], 'nmc7':[], 'nmc8':[], 'nmE7':[], 'nmE8':[],
'ors2':[], 'ors5':[], 'ors6':[], 'ors7':[], 'ucct':[], 'Ucct':[],
'or15':[], 'or17':[], 'or19':[], 'or20':[], 'or21':[], 'or22':[],
'date':[], 'dat2':[], 'dat6':[], 'wkdy':[], 'bshr':[], 'hldy':[],
'wkds':[], 'wkdo':[], 'mnts':[], 'mnto':[],
'yea2':[], 'mnt2':[], 'mnt6':[], 'day2':[], 'day5':[],
'hrs2':[], 'hrs4':[], 'min2':[], 'min4':[], 'scn2':[], 'DPrt':[],
'DPnb':[], 'DPmm':[], 'DPbn':[], 'DPod':[], 'DP10':[], 'DPoh':[],
'excl':[], 'exc2':[], 'exc3':[], 'exc4':[], 'exc5':[], 'exc6':[],
'null':[], 'copy':[], 'shfl':[], 'eval':[], 'ptfm':[]},
assignparam = {'default_assignparam' :
          {'(category)' : {'(parameter)' : 42}},
          '(category)' : {'(column)' : {'(parameter)' : 42}}},
assigninfill = {'stdrdinfill':[], 'MLinfill':[],
          'zeroinfill':[], 'oneinfill':[],
          'adjinfill':[], 'meaninfill':[], 'medianinfill':[],
          'modeinfill':[], 'lcinfill':[], 'naninfill':[]},
assignnan = {'categories':{}, 'columns':{}, 'global':[]},
transformdict={}, processdict={}, evalcat=False,
privacy_encode = False, printstatus=True)
```

Or for postmunge(.) with full range of parameters:

```
test, testID, testlabels, \
labelsencoding_dict, postreports_dict = \
am.postmunge(postprocess_dict, df_test,
 testID_column = False, labelscolumn = False,
 pandasoutput = False, printstatus = True,
 TrainLabelFreqLevel = False, featureeval = False,
 driftreport = False,
 LabelSmoothing = False, LSfit = False, inversion = False,
 traindata = False,
 returnedsets = True, shuffletrain = False)
```

## C  ASSIGNING TRANSFORMS AND INFILL

Assigning root categories is conducted in assigncat parameter, assigning infill in assigninfill, and parameters in assignparam - e.g. for a train set df_train with column headers 'col1' and 'col2' we could assign string parsing (splt) and a string parse family tree (or19) with infill types zero infill and ML infill.

Since 'splt' transform accepts parameters, we'll also demonstrate passing parameter of excluding space and special characters, such as to promote single word overlap detections.

Note any columns we don't explicitly assign will defer to automation or we could turn off automated defaults for pass-through of other columns by passing automunge parameter powertransform = 'excl'.

```
train, trainID, labels, \
validation1, validationID1, validationlabels1, \
validation2, validationID2, validationlabels2, \
test, testID, testlabels, \
labelsencoding_dict, finalcolumns_train, finalcolumns_test, \
featureimportance, postprocess_dict \
= am.automunge(df_train,
  assigncat = {'splt':['col1'], 'or19':['col2']},
  assigninfill = {'zeroinfill':['col1'], 'MLinfill':['col2']},
  assignparam = {'splt' : {'col1' :
                 {'space_and_punctuation' : False}}})
```

# D    OVERWRITING SETS OF TRANSFORMS

Here we'll demonstrate overwriting sets of transformations with the transformdict parameter passed to automunge(.).

Let's use the example from paper of 'or19' application of 'nmr7' to extract numerical portions which has result of returning the zip codes in the address examples. We noted in paper that there might be benefit to encoding the output as categorical instead of a numeric (z-score) normalization, so let's demonstrate overwriting the pre-defined transform to apply a categoric encoding downstream of the 'nmr7'.

Referring to Figure 7, we see the relevant family tree is for category 'nmc8' which has a children primitive entry of 'nmbr'. These family trees are also available for inspection in the READ ME.

So if we want to instead encode the numeric extract as categorical, we could overwrite 'nmc8' in a custom passed transformdict.

Note that since 'nmc8' is called as part of an offspring generation, the 'nmc8' upstream primitives (parents / siblings / auntsuncles / cousins) aren't inspected in context of 'or19', so we only have to worry about the downstream primitives (children / niecesnephews / coworkers / friends).

Let's demonstrate the overwrite, we'll instead use 'ord3' for ordinal encoding by frequency.

```
transformdict = {'nmc8' : {'parents' : ['nmc8'],
                           'siblings' : [],
                           'auntsuncles' : [],
                           'cousins' : ['NArw'],
                           'children' : ['ord3'],
                           'niecesnephews' : [],
                           'coworkers' : [],
                           'friends' : []}}
```

And since we are using existing categories from the library, we don't need to repopulate a corresponding processdict.

Then when we call automunge(.) we can pass this transformdict to overwrite 'nmc8':

```
train, trainID, labels, \
validation1, validationID1, validationlabels1, \
validation2, validationID2, validationlabels2, \
test, testID, testlabels, \
labelsencoding_dict, finalcolumns_train, finalcolumns_test, \
featureimportance, postprocess_dict \
= am.automunge(df_train,
  assigncat = {'or19':['col2']},
  transformdict = transformdict)
```

The returned columns log the steps of transformations via suffix appenders, so this replaces the original returned column 'col2_UPCS_nmc7_nmbr' with 'col2_UPCS_nmc7_ord3'.

# E    FEATURE IMPORTANCE

The Automunge library includes an option for push-button feature importance evaluation by shuffle permutation, in which a model is trained on the full training set feature set with a base accuracy evaluated on a partitioned validation set. Metrics are then derived by shuffling the rows of a target feature to determine the resulting delta from dampened accuracy. In the first metric all of the columns derived from the same source feature are shuffled, and a higher metric signals higher influence of the source feature. In the second metric, all columns except for a target column originating from the same source feature are shuffled, and a lower metric signals higher relative influence in comparison to the other columns derived from the same source feature. The current basis for performance metrics is accuracy for classification and mean squared log error for regression. Note that the method requires the inclusion and designation of a label column by the labels_column parameter.

Feature importance is supported by the following automunge(.) parameters:

- featureselection: boolean (default False), when True feature importance is performed
- featuremethod: accepts entries {'default', 'pct', 'metric', 'report'}, where 'default' evaluates feature importance and then processes data as in a general automunge(.) call, 'report' evaluates feature importance without further processing of data, and 'pct' or 'metric' activate a dimensionality reduction on returned data based on the results, further detailed in READ ME documentation.

The results of an evaluation are returned in the printouts, and then also in the returned dictionary featureimportance. Sorted results are further available in the returned postprocess_dict dictionary under the key 'FS_sported'.

Note that feature importance can also be conducted on subsequent data in the postmunge(.) function by activating the boolean featureeval parameter, with results available in printouts and returned in the returned dictionary postreports_dict.

The second metric for relative importance between columns derived from the same feature are particularly useful for evaluating influence of different segments of a feature set's distribution. For example, when a categoric feature is encoded by 'text' one-hot encoding the metrics may indicate which of the categoric entries are more influential to the model. Similarly, when a numeric feature is encoded by 'tlbn' tail bins encoding, the metrics may indicate which segments of the numeric set's distribution are more influential to the model.

Important to keep in mind that feature importance metrics are as much a measure of the model as they are of the features.

## F    A FEW HELPFUL HINTS

A few highlights that might make things easier for first-timers:

1) automunge(.) returns sets as numpy arrays by default (for universal compatibility with ML platforms). A user can instead receive the returned sets as pandas dataframes by passing the parameter pandasoutput = True

2) Even if the sets are returned as numpy arrays, you can still inspect the returned column headers with the returned list we demonstrate as finalcolumns_train

3) Printouts are turned on by default, they can be turned off with printstatus=False

4) Note for data sets with just a few rows, such as those demonstrated here, there is a PCA heuristic to apply dimensionality reduction when the number of features is more than 50% of the number of observations in the train set (this is a somewhat arbitrary heuristic). This can be turned off with ML_cmnd = {'PCA_type':'off'}.

5) Speaking of PCA, if you do want to apply PCA, a useful option allows you to exclude from dimensionality reduction boolean or ordinal encoded columns, available with ML_cmnd = {'PCA_cmnd':{'bool_ordl_PCAexcl':True}}.

6) Note that data shuffling is on by default for the train set and off by default for the test sets returned from automunge(.) and postmunge(.). If you want to shuffle the test data in automunge too you can pass shuffletrain = 'traintest'. Or to shuffle the test data returned from postmunge you can pass the postmunge parameter shuffletrain = True.

7) The automated feature importance evaluation is easy to use, you just need to be sure to designate a label column with labels_column = 'column_header'. Then just pass featureselection = True and printouts will return results as well as the returned report featureimportance

8) To ensure that you can later prepare additional data for inference, please be sure to save the returned postprocess_dict such as with pickle library.

9) Importantly, when you pass a train set to automunge(.) please designate any included labels column with labels_column = (label column header string), which may be an integer index for numpy arrays. When you go to process additional data in postmunge, the columns must have consistent headers as those originally passed to automunge. Or if you originally passed numpy arrays, just be sure that the columns are in the right order. If you're passing postmunge(.) data that includes a column originally designated as a label to automunge, just apply labelscolumn = True.

10) Speaking of numpy arrays, if you pass numpy arrays instead of pandas dataframes to the function, all of the column assignments will accept integers for the column number.

11) When applying ML infill, which is based on Scikit-Learn Random Forest implementations, a useful ML_cmnd if you don't mind a little more training time is to increase the number of estimators as e.g. ML_cmnd = {'MLinfill_cmnd':{'RandomForestClassifier':{'n_estimators':1000}, 'RandomForestRegressor':{'n_estimators':1000}}}

12) Note that any columns you want to exclude from processing, you can either assign them to root category 'excl' in assigncat if you don't mind their retention (noting that they will still be included in ML infill so will need numerical encoding), or you can carve them out from the set to be returned in ID sets consistently shuffled and partitioned such as between training and validation data, just pass a list of column headers to trainID_column and/or testID_column. You can also turn off the automated transforms and only perform those designated in assigncat by passing powertransform='excl'

## G    INTELLECTUAL PROPERTY DISCLAIMER

