# OpenReview forum: "Parsed Categoric Encodings with Automunge"
_ICLR.cc/2021/Conference — Reject_

### Official Review · AnonReviewer2 · 2020-10-27
**The authors present a python package to process data in the form of strings.**

**Rating:** 6
**Confidence:** 2

**Review:**

This package aims at automating some repetitive tasks that data analysts (especially within the NLP domain) deal with. I confirm that tasks like feature selection and string encoding are beneficial for applied researchers with textual data. I can see a wide interest in the community.
On the negative side, the paper lacks a literature review and comparison with any existing and relevant work. I expected to see performance plots for different tasks (whether just for this package or with comparison to some baseline alternatives.

As a minor comment, I don't think having "String theory" in the title is a good idea because this keyword is already taken to refer to another scientific topic ( a sub-field of physics).

---

> ### Author Response · Authors · 2020-11-11
> **Further clarifications**
>
> Thank you reviewer for your comments on this work. I appreciate your recognition of the benefit of automating repetitive tasks that data analysts, especially in NLP domain, deal with. Thank you for confirming that feature selection and string encoding will be beneficial for applied researchers with textual data.
>
> I hope you are right about potential for a wide interest in the community, one of the consistent challenges for the library developers has been getting the word out, as Automunge is an open source library without resources for “public relations” or advertising. I believe that this library could be of broad benefit to both machine learning researchers and practitioners, as tabular data preprocessing conventions have yet to hone in on a single mainstream standard.
>
> With regards to a literature review and comparison to existing and relevant work, this is partly a result of research being conducted by invention and building things from scratch (primarily on top of the Pandas library). The seed of this project originated from some beginner tabular data competitions on Kaggle, and the developers basically took the approach of building what was seen as an unmet need for a tabular data standard from the ground up. The development process has been incremental and evolutionary, and along the way I believe some material improvements have been incorporated over what is otherwise available in mainstream practice for tabular data preprocessing.
>
> With respect to performance plots for different tasks, I am taking this as a good idea for further research. One of the drivers for various design decisions has been speed of application, particularly for subsequent data processed in the postmunge(.) function. Part of the challenge for benchmarking purposes is budgetary constraints associated with expensive licenses for commercial alternatives. We did include some benchmarking for speed in the Jupyter notebooks uploaded with the supplemental material (see for example the uploaded notebook “efficiency_tests_061020.ipynb”).
>
> I believe one of the key advantage of this framework over a mainstream option like scikit-learn to originate from the simplicity of populating a single dictionary “fit” to properties of a training set which can be shared and published by researchers for fully consistent processing of additional data - as may benefit reproducibility of benchmarks and experiments. Other aspects of novelty such as these string parsing methods and automated ML for missing data infill are also a material improvement.
>
> With regards to the use of the term “string theory”, I hope you will grant this author a small indulgence. It’s use is admittedly a little on the humorous side, we justified that based on an assumption that there was not likely to be any confusion about the overlap between these very very different domains, especially in the context of the phrase “Parsed Categoric Encodings”. If you consider it beneficial for moving forward we would certainly be willing to strike those two words from the title.
>
> Again I certainly appreciate your review and recognition of the potential benefit of this library to applied research. I have also publicly responded to the other reviewers, and have taken an action to upload an additional validation demonstration notebook which should follow in the next few days. Best regards.

---

> > ### Author Response · Authors · 2020-11-13
> > **validation notebook now uploaded**
> >
> > Hello. I noted in my response that I would upload an additional validation notebook. That notebook is now uploaded in the supplemental material under the folder 'stringparse_validation_notebook_1112'.
> >
> > Best regards.

---

> ### Author Response · Authors · 2020-11-14
> **new version uploaded**
>
> Hello reviewers. I just uploaded a new revision attempting to address a few comments raised in your discussions. The revisions are as follows:
>
> 1) Changed the title at the suggestion of one of reviewers
> 2) Added a small clarification to Section 5 regarding family tree primitives.
> 3) Replaced Figure 7 with a cleaner and more comprehensive version for clarity.
> 4) Added a new Experiments section in what is now Section 7. These experiments are meant to address comments raised by multiple reviewers that the work would be more aligned with ICLR to demonstrate how string parsing improves model accuracy. The section also introduces the Automunge feature importance method which relies on auto ML models evaluated by shuffle permutation. The experiments compared between standard tabular categoric representation techniques and parsed categoric encodings and demonstrated a material improvement to model accuracy as well as feature importance as a result of string parsing aggregations. The experiments also demonstrated some informative benchmarks in comparison of traditional categoric encodings for ML.
> 5) Added and revised some of the Discussions section in what is now Section 8, including a clarification that the family tree primitives represent a scientifically novel framework and that the automation of string parsing for categoric encodings is a novel invention.
> Added a demonstration notebook to the Supplemental Material corresponding to the Experiments section. Also replaced the code repo from version 4.88 to the current version 5.22.
>
> I appreciate all of the comments that you have shared. I hope that the new experiments in particular demonstrate the relevancy of automated string parsed categoric encodings to ICLR. Any additional comments are welcome and I appreciate the chance to share this work.
>
> Best regards

---

### Official Review · AnonReviewer4 · 2020-11-01
**Very nice featurization library --- not a topic for ICLR**

**Rating:** 4
**Confidence:** 4

**Review:**

The submitted paper describes a very nice featurization library, AutoMunge, that converts NLP into features suitable for NNs.

It's clear that the authors of the library have put a lot of thought into its construction, and it looks very useful.

However, ICLR is about /learning/ representations, not about feature engineering. So this is off-topic for the conference. To make it on-topic, the authors could, e.g., compare using standard word representation techniques vs AutoMunge on a set of NLP tasks using some popular modern NLP architecture (perhaps BERT?). That would be a really interesting paper.

---

> ### Author Response · Authors · 2020-11-11
> **ICLR applicability**
>
> Thank you reviewer for your interpretation of this work. I am reading the primary consideration of your review to be based on the premise that Automunge is off-topic for the ICLR conference. I will present here a short summary of why I do not believe that to be the case.
>
> One of the papers we cited was “Efficient Estimation of Word Representations in Vector Space” by Mikolov et al from ICLR 2013, which rolled out the Word2Vec method for vocabulary vectorization, a precurser to the types of contextual vocabulary embeddings applied in NLP applications like GPT-3. Quoting Gary Marcus in his recent book Rebooting AI, “Rumors of the replacement of feature engineering have been somewhat exaggerated; the hard work that goes into crafting representations like Word2Vec still counts as feature engineering, just of a different sort…”. Even if string parsing is a type of feature engineering, so is Word2Vec. More particularly, we noted in our paper that vocabulary vectorization of the like of Word2Vec is inaccessible for certain types of categoric features that may be found in a tabular data set. We gave two examples of addresses and serial numbers in which language models trained on public text corpus may be insufficient for application to esoteric domains, especially considering that in tabular data sets categoric feature sets lack the context of a surrounding text corpus that may enable a fine-tuning of a language model. These type of scenarios are not uncommon in real-world tabular data sets.
>
> The premise of our work is that in a tabular data application, models will often benefit by improving the information retention of categoric encodings by extracting grammatical structure shared between categoric entries as opposed to coarse-grained encodings of mainstream methods.
>
> We believe that the string parsing operation to be beneficial both as a general purpose supplement to categoric encodings to less advanced users, but also useful for sophisticated users who might otherwise consider applying advanced NLP methods like BERT to tabular applications who recognize that there are some types of esoteric domains where BERT may not be as viable. Even for next generation technologies we are seeing from the likes of GPT-3, having a formal framework for tabular data processing will still be necessary - if we have a NLP model that can write software in python, that doesn’t mean we don’t need python. Thus Automunge is infrastructure that could be built on top of with NLP applications. A foundation.
>
> I appreciate your review of this work. I hope you might consider reading my responses to the other reviewers as well before making your final decision. Best regards.

---

> > ### Author Response · Authors · 2020-11-13
> > **additional validation notebook now uploaded**
> >
> > Hello. Wanted to quickly share that I just uploaded an additional validation notebook in the supplemental material per the suggestion of one of the other reviewers. That notebook is available in the folder 'stringparse_validation_notebook_1112'.
> >
> > Best regards.

---

> > ### Comment · AnonReviewer4 · 2020-11-24
> > **Thanks for testing with random forests**
> >
> > Thanks to the authors for responding to the review by testing with a random forest on a Kaggle competition. The accuracy lift by automunge seemed very small. My original suggestion was to try a more up-to-date NLP model: that kind of model may be able to exploit the feature structure produced by automunge.
> >
> > Due to the (somewhat inconclusive) experiment, I've raised my score to 4. Again, the software looks very useful, but the paper hasn't passed the bar for ICLR.

---

> > > ### Author Response · Authors · 2020-11-24
> > > **Small improvements for fundamentals**
> > >
> > > Thank you for your response. While the accuracy impact for this single feature example was small, this was just a representative example, and we noted that results will vary based on data set properties. The question of categoric encoding for tabular data is one of most fundamental aspects a model. Even a small improvement may in aggregate have a big impact on industry. The point is that we have automated the recovery of information that is otherwise not presented to the model in mainstream tabular learning frameworks.
> > >
> > > Best regards.

---

> ### Author Response · Authors · 2020-11-14
> **new version uploaded**
>
> Hello reviewers. I just uploaded a new revision attempting to address a few comments raised in your discussions. The revisions are as follows:
>
> 1) Changed the title at the suggestion of one of reviewers
> 2) Added a small clarification to Section 5 regarding family tree primitives.
> 3) Replaced Figure 7 with a cleaner and more comprehensive version for clarity.
> 4) Added a new Experiments section in what is now Section 7. These experiments are meant to address comments raised by multiple reviewers that the work would be more aligned with ICLR to demonstrate how string parsing improves model accuracy. The section also introduces the Automunge feature importance method which relies on auto ML models evaluated by shuffle permutation. The experiments compared between standard tabular categoric representation techniques and parsed categoric encodings and demonstrated a material improvement to model accuracy as well as feature importance as a result of string parsing aggregations. The experiments also demonstrated some informative benchmarks in comparison of traditional categoric encodings for ML.
> 5) Added and revised some of the Discussions section in what is now Section 8, including a clarification that the family tree primitives represent a scientifically novel framework and that the automation of string parsing for categoric encodings is a novel invention.
> Added a demonstration notebook to the Supplemental Material corresponding to the Experiments section. Also replaced the code repo from version 4.88 to the current version 5.22.
>
> I appreciate all of the comments that you have shared. I hope that the new experiments in particular demonstrate the relevancy of automated string parsed categoric encodings to ICLR. Any additional comments are welcome and I appreciate the chance to share this work.
>
> Best regards

---

### Official Review · AnonReviewer1 · 2020-11-02
**Useful software but insufficient scientific contribution**

**Rating:** 4
**Confidence:** 5

**Review:**

This paper introduces a library that preprocesses tabular data called Automunge.

For software packages I feel that one of two criteria must be met: The software implements a scientifically novel algorithm, framework, model, etc.; or the software package is so complex that a well-designed implementation in itself is of scientific significance.

Whereas Automunge seems like a useful library, I am not convinced that it falls in either of these two categories. As such, I am not sure it justifies a publication at a machine learning conference. I suggest the authors target a different venue (e.g., PyCon, SysML, etc.) or elaborate on the scientific impact of their software (e.g., show experimentally that this framework allows practitioners to train better performing models).

Pros

* The library seems useful

Cons

* No significant novelty
* Little relevance to the scientific machine learning community
* Not very clearly written

---

> ### Author Response · Authors · 2020-11-11
> **Contributions**
>
> I appreciate that you offered two specific criteria for software packages, I believe this software has met both of these criteria as follows:
>
> Criteria one: “The software implements a scientifically novel algorithm, framework, model, etc.”
> I believe the family tree primitives as described in Figure 6 meet this criteria, for the reason that they have formalized a fundamental aspect of processing tabular data, as enabling a simple means for command line specification of multi-transform sets that may include generations and branches of derivations. I believe the family tree primitives to be somewhat fundamental, and any data transformation sets as may be applied to a single feature set of origination (as would be found in a “tidy data” set) can be universally expressed by way of these simple and novel primitives applied by way of recursion.
>
> More particularly for string parsing considerations, the paper does not just introduce string parsing, it offers a comprehensive overview of the various permutations that may be applied for this purpose. We sought for this treatment of string parsing methods to be exhaustive. We believe that string parsing is appropriate for a machine learning conference because of just how fundamental is the application for tabular data applications of machine learning, which in practice is generally comprised of just two broad categories of feature set types - numeric and categoric. We have introduced a novel automated approach for encoding tabular categoric features.
>
> Although the benefit of string parsing is expected to vary based on esoteric characteristics of target feature sets, the paper operated on the premise of a self-evident benefit to ML for improved information retention of extracting grammatical structure that may be shared between categoric entries for presentation to a training operation in comparison to coarse-grained representations. That being said I am working now on an additional demonstration jupyter notebook to be uploaded to the supplemental material and will advise when it is ready in which I intend to experimentally demonstrate the benefit as you suggested.
>
> Criteria two: “the software package is so complex that a well-designed implementation in itself is of scientific significance.”
> I believe the simplicity of the package is deceptive for the amount of complexity that is abstracted away. I recently attended a data science conference at a high profile university where the keynote speaker described a project to apply machine learning to predict missing data infill for a specific tabular data application in industry. Automunge offers a generalized solution and abstracts away all of the complexities for any tabular data application. It is a push-button autoML solution for missing data infill, and all of the string parsing methods demonstrated have built in support.
>
> One of the most useful abstractions for purposes of hiding complexity is the manner in which the application of automunge(.) populates a python dictionary “fit” to properties of the train set, capturing all of the steps and parameters of transformations, such that for subsequent data, including streams of data for inference, consistent preparations may be applied quickly and efficiently in the postmunge(.) function with only the prerequisite of passing this dictionary. This practice of basing properties of transformations explicitly on properties from a designated train set is an improvement on what is still common in mainstream practice to normalize train/test/validation sets separately - which introduces issues of potential stochastic inconsistency and data leakage. We noted too in the Broader Impacts appendix that the ability of researchers to publish these populated dictionaries could benefit reproducibility of benchmarks and experiments.
>
> Thank you for the recognition that you believe this library would be useful. That is our goal.
>
> With respect the the “cons” that you noted:
>
> Regarding novelty: we believe the push button automation of string parsing operations to be novel. We believe the integration of command line specification for multi-transform sets, autoML missing data infill, and various other features of the library to be novel. We believe the family tree primitives to be novel and a particularly useful fundamental reframing of specifying transformation sets via recursion.
>
> I hope you will forgive my writing style, you noted that it was not clearly written. Partly this was associated with trying to cover a lot of ground I suspect. I believe the family tree primitives description benefits by taking into account the demonstrations of Figures 4, 5, 6, and 7 which illustrate their application in practice for the given ‘or19’ root category example.
>
> Thank you again for your review. Happy to answer further questions. I hope you might reconsider your rating based on this feedback. If you are unsure please consider reviewing my response to the other two reviewers for context. Best regards.

---

> > ### Author Response · Authors · 2020-11-13
> > **validation notebook now uploaded**
> >
> > Hello. I noted in my response that I would upload an additional validation notebook as per your suggestion. That notebook is now uploaded in the supplemental material under the folder 'stringparse_validation_notebook_1112'.
> >
> > Best regards.

---

> ### Author Response · Authors · 2020-11-14
> **New version uploaded**
>
> Hello reviewers. I just uploaded a new revision attempting to address a few comments raised in your discussions. The revisions are as follows:
>
> 1) Changed the title at the suggestion of one of reviewers
> 2) Added a small clarification to Section 5 regarding family tree primitives.
> 2) Replaced Figure 7 with a cleaner and more comprehensive version for clarity.
> 3) Added a new Experiments section in what is now Section 7. These experiments are meant to address comments raised by multiple reviewers that the work would be more aligned with ICLR to demonstrate how string parsing improves model accuracy. The section also introduces the Automunge feature importance method which relies on auto ML models evaluated by shuffle permutation. The experiments compared between standard tabular categoric representation techniques and parsed categoric encodings and demonstrated a material improvement to model accuracy as well as feature importance as a result of string parsing aggregations. The experiments also demonstrated some informative benchmarks in comparison of traditional categoric encodings for ML.
> 4) Added and revised some of the Discussions section in what is now Section 8, including a clarification that the family tree primitives represent a scientifically novel framework and that the automation of string parsing for categoric encodings is a novel invention.
> 5) Added a demonstration notebook to the Supplemental Material corresponding to the Experiments section. Also replaced the code repo from version 4.88 to the current version 5.22.
>
> I appreciate all of the comments that you have shared. I hope that the new experiments in particular demonstrate the relevancy of automated string parsed categoric encodings to ICLR. Any additional comments are welcome and I appreciate the chance to share this work.
>
> Best regards

---

### Author Response · Authors · 2020-11-17
**New Appendix E added for Feature Importance**

Hello reviewers. Since the new Experiments section makes use of the Automunge Feature Importance methods, I decided to add a new appendix to detail how to apply, uploaded now as Appendix E - Feature Importance.

I also updated the Jupyter notebook for demonstrating appendix code that was included in Supplemental Material ('appendix_code_validation_111620.ipynb') to include some demonstrations of feature importance implementation at the conclusion. Best regards.

---

> ### Author Response · Authors · 2020-11-19
> **Small revision to align with revised Figure 7**
>
> Hi reviewers, as an update on progress, I just uploaded a new version which includes a small revision to the paragraph preceding Figure 7, partly to better align with the more comprehensive version of the Figure, also for improved clarity. Best regards.

---

### Comment · ~Nicholas_Teague1 · 2021-01-14
**Final response**

Two closing thoughts:

1. We believe the parsed categoric encodings represent a improved alternative one-hot encoding for increased information retention in presentation to a learning algorithm. These encodings translate categoric entries to a vectorized representation as a function of shared grammatical structure between entries - structure that would otherwise be hidden from model training.

2. We believe the family tree primitives are an important contribution, representing a fundamental reframing of command line specification for transformation sets that may include generations and branches of derivations applied by recursion. We were disappointed that the reviewers did not seem to get this point.

Please feel free to visit the Automunge GitHub for full documentation. We welcome further feedback or requests.

Best regards.

---

### Decision · Program_Chairs · 2021-01-07
**Final Decision**

**Decision:**

Reject

**Comment:**


This paper presents "Automunge" a python library for pre-processing tabular data.
The authors develop a useful library that can be used by practicioners for data engineering in NNs applications.
The reviewers raised a common concern regarding the lack of focus on the actual usefulness of the librabry in improving the
performance of the models that is applied on. A common concern was the lack of performance plots compared to other alternatives.
In the response the authors have done a rather thorough job of addressing the reviewers comments and
adding material in the supplementary. However, given the current presentation, the manuscript needs a considerable amount of  rewriting to incorporate the suggested changes into the main paper. As it is, I don't think ICLR is the right venue for the manuscript.  It might reach its audience better in venues like SysMl or PyCon also suggested by a reviewer.